# Real3D-AD: A Dataset of Point Cloud Anomaly Detection

**Jiaqi Liu**[1,*] **Guoyang Xie**[1,2,*], **Ruitao Chen**[1,*], **Xinpeng Li**[1], **Jinbao Wang**[1,†], **Yong Liu**[3],
**Chengjie Wang**[3,4], **Feng Zheng**[1,†]
[1]Department of Computer Science and Engineering,
Southern University of Science and Technology, Shenzhen 518055, China
[2]NICE Group, University of Surrey, Guildford, GU2 7YX, United Kingdom
[3]Tencent Youtu Lab, Shenzhen 518057, China
[4]Shanghai Jiao Tong University, Shanghai 200240, China
liujq32021@mail.sustech.edu.cn, guoyang.xie@ieee.org
linkingring@163.com, chenrt2022@mail.sustech.edu.cn
li.xin.peng@outlook.com, chaosliu@tencent.com
jasoncjwang@tencent.com, zfeng02@gmail.com

## Abstract

High-precision point cloud anomaly detection is the gold standard for identifying the defects of advancing machining and precision manufacturing. Despite some methodological advances in this area, the scarcity of datasets and the lack of a systematic benchmark hinder its development. We introduce Real3D-AD, a challenging high-precision point cloud anomaly detection dataset, addressing the limitations in the field. With 1,254 high-resolution 3D items (from forty thousand to millions of points for each item), Real3D-AD is the largest dataset for high-precision 3D industrial anomaly detection to date. Real3D-AD surpasses existing 3D anomaly detection datasets available regarding point cloud resolution (0.0010mm-0.0015mm), 360 degree coverage and perfect prototype. Additionally, we present a comprehensive benchmark for Real3D-AD, revealing the absence of baseline methods for high-precision point cloud anomaly detection. To address this, we propose Reg3D-AD, a registration-based 3D anomaly detection method incorporating a novel feature memory bank that preserves local and global representations. Extensive experiments on the Real3D-AD dataset highlight the effectiveness of Reg3D-AD. For reproducibility and accessibility, we provide the Real3D-AD dataset, benchmark source code, and Reg3D-AD on our website:https://github.com/M-3LAB/Real3D-AD.

## 1 Introduction

**Real 3D-AD Motivation: 3D > 2.5D.** There is a solid need to propose a high-resolution point cloud anomaly detection dataset to tap the gap between the academy and industry, which brings the capabilities of point cloud anomaly detection to the factory floor. Point cloud anomaly detection is widely deployed in real-world production lines. However, 3D anomaly detection datasets released in the academy are RGBD (2.5D), which is outside the demand of industrial manufacturing. Advanced machining and precision manufacturing require no blind spots throughout the inspection process. Nevertheless, the blind spots exist because RGBD datasets are achieved via single-view scanning. The lack of a real point cloud anomaly detection dataset hinders the further development of 3D

---

[*]Contributed Equally, [†]Corresponding Authors.

37th Conference on Neural Information Processing Systems (NeurIPS 2023) Track on Datasets and Benchmarks.

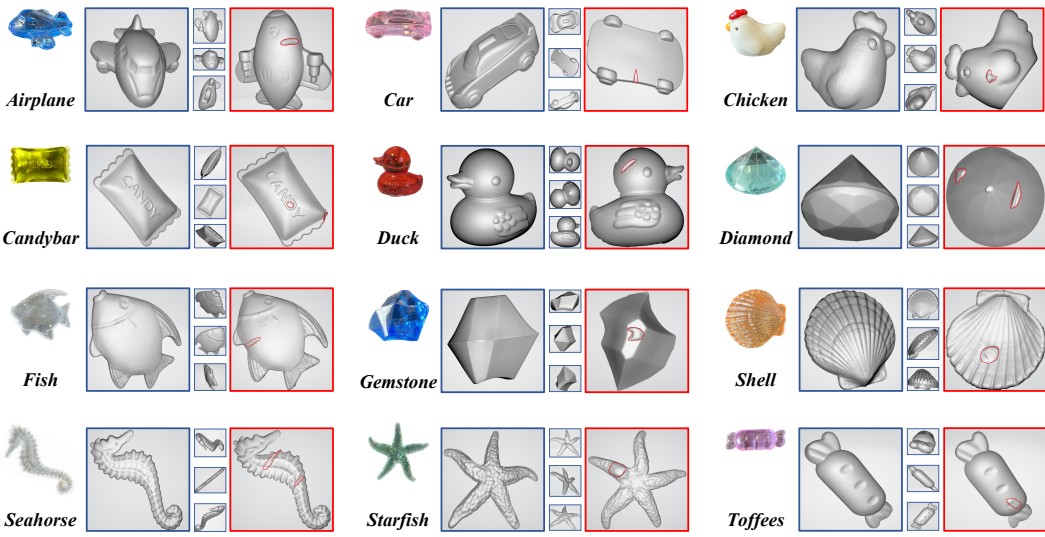

Figure 1: Real3D-AD dataset examples for each category. The blue box indicates the normal images in the training dataset. The red box denotes the abnormal images in the test dataset. There are no blind spots in Real3D-AD since our dataset are achieved by scanning all the views of the object instead of the single view photoed by RGBD camera.

anomaly detection. Because of this, it is crucial and urgent to propose a point cloud anomaly detection dataset that meets industrial manufacturing needs.

**Limitation of current 3D-AD dataset and Real3D-AD advantage.** To address this problem, we present a large-scale, high-resolution 3D anomaly detection dataset, Real3D-AD, to support the research and development of 3D anomaly detection methods. Although two 3D anomaly detection dataset (MVTec 3D-AD [20] and Eyescandies [3]) has been proposed, there are still several limitations: 1) the precision of MVTec 3D-AD are insufficient to satisfy the requirements of high-precision point cloud anomaly detection. Specifically, MVTec 3D-AD offers a limited number of 4,147 point clouds per object with a point precision of 0.11mm. Real3D-AD offers a significantly greater number of point clouds per object, estimated at 1.3 million, compared to MVTec 3D-AD, which is approximately 100 times larger. Moreover, the point precision of Real3D-AD reaches up to 0.010mm, which is ten times more advanced than MVTec 3D-AD. The detailed analysis is given in Table 1. 2) There are blind spots of 3D anomaly detection datasets if adopting an RGBD camera to collect the 3D data, like MVTec 3D and Eyescandies. Identifying defects may pose a challenge when relying solely on a single view for inspection. Real3D-AD is collected using high-resolution laser scans, perfect for spotting the product's defects everywhere, as shown in Figure 1. 3) The simulated dataset (Eyescandies) makes it hard to extend the realistic scenario. Due to the commercial privacy of real products, it is difficult to collect the CAD model of real-world products. So most researchers adopt the simulated software, like *Blender* framework [9]. Nevertheless, both synthesizing texture and anomaly details are not achieved with high fidelity. Real3D-AD collects the products from real-world applications and obtains excellent prototypes via the high-resolution 3D scanner. Hence, as shown in Table 3, we can conclude that three key characteristics distinguish Real3D-AD relative to prior work on the 3D anomaly detection dataset: high precision, no blind spot, and realistic high-accuracy prototype.

**Benchmark & Baseline.** To expedite research efforts towards developing a universal high-precision point cloud anomaly detection approach, we have constructed a comprehensive and structured large-scale benchmark, referred to ADBENCH-3D. Additionally, we have developed a registration-based baseline method that aligns with the prerequisites of high-resolution 3D anomaly detection. Given the practical constraints, the number of training datasets available for each category is restricted (less than or equal to four), as creating a high-accuracy prototype for each category is a time-intensive process (requiring up to two days per category). The configuration of ADBENCH-3D varies from contemporary unsupervised anomaly detection tasks in the realm of 3D. Section 4.1 provides a

comprehensive description of the setting. To be more specific, the training examples are limited ($\leq$ 4), and the test samples are only scanned by one side. The motivation is to simulate the real-world application: The scanning positions on the production line are fixed, and one position can only scan the results of one product side. Furthermore, to facilitate precise performance comparison within the community and guarantee replicability, the ADBENCH-3D framework encompasses a comprehensive end-to-end pipeline that includes data preprocessing, 3D-AD algorithms, evaluation scripts, metrics, and a visualization toolkit. ADBENCH-3D comprises a set of 8 fundamental 3D anomaly detection methodologies that have been implemented and tested on the Real3D-AD dataset. Moreover, the results presented in Table 4 indicate that the majority of current 3D-AD techniques are unable to attain satisfactory performance in Real3D-AD, as evidenced by their object-level AUROC score falling short of 50%. Consequently, we propose a registration-based 3D-AD method (Reg3D-AD) that serves as a versatile solution to cater to the requirements of the Real3D-AD dataset. The Rege3D-AD model introduces a novel feature memory bank, as illustrated in Figure 8, designed to preserve both the local and global features. The test objects are aligned with the training prototype during the inference process, and their features are extracted locally and globally. The defects are identified by assessing the distance between the features of the test object and the training prototypes. Therefore, Real3D-AD and ADBENCH-3D present a step towards unifying disjoint efforts in 3D anomaly detection research and pave the way toward a deeper understanding of 3D anomaly detection models.

Overall, the main contributions of this paper are:

- We create the first-ever high-resolution 3D anomaly detection dataset (Real3D-AD), enabling the design of high-resolution 3D anomaly detection algorithms and applying it to publicly available. Real3D-AD exhibits three primary attributes that set it apart from previous studies on 3D anomaly detection datasets. These attributes include a high level of precision, an absence of blind spots, and a realistic, high-accuracy prototype.

- The end-to-end pipeline offered by ADBENCH-3D includes data preparation, data splits, evaluation metrics and scripts, and visualization toolkits. ADBENCH-3D conducts a large-scale systematic assessment (8 main algorithms on Real3D-AD).

- We propose a general-purposed registration-based 3D anomaly detection method (Reg3D-AD). The efficacy of Reg3D-AD has been demonstrated through comprehensive experimentation on the Real3D-AD dataset, surpassing the performance of the subsequent most effective approach by a significant degree.

## 2 Related work

**3D-AD Datasets.** The datasets for 2D anomaly detection (2D-AD) are abundant, with a history tracing back to 2007 [29]. Over 20 different datasets are available for 2D-AD [11, 25, 17, 31]. The numerous 2D-AD datasets have given rise to many related works. Some studies have approached it from the perspective of image reconstruction [35, 8], feature distillation [12, 4, 27], and feature comparison [22]. A body of research also focuses on specific scenarios such as few-shot anomaly detection [30, 36, 7] and noisy anomaly detection [16]. In contrast, the number of datasets for 3D anomaly detection (3D-AD) is considerably limited. The first 3D-AD dataset was introduced in 2021, and today, there are only two 3D-AD datasets, MVTec 3D-AD dataset [20] and Eyecandies dataset [3]. MVTec 3D-AD [20] is a new dataset designed for 3D point cloud anomaly detection, and it is the only point cloud dataset for AD. It contains 2,656 pairs of images as training sets, 294 pairs of images as validation sets, 249 pairs of normal images, and 948 pairs of abnormal images to form a test set. The dataset comprises a total of 41 distinct types of anomalies, with a combined count of 1148 anomaly regions. Each pair of images consists of RGB images and tiff images representing the spatial coordinates of each pixel. The resolution of the images varies from $400 \times 400$ to $900 \times 900$. The Eyecandies dataset [3] is a novel synthetic dataset comprising ten different categories of candies rendered in a controlled environment. It consists of 13,250 pairs of normal samples and 2,250 pairs of abnormal samples. Each depth image corresponds to six RGB images under different lighting conditions. Both MVTec 3D-AD and Eyecandies are RGBD datasets, limited to single-view information. To further explore the value of spatial information in the AD task, we propose the Real 3D-AD dataset, which expands the object information to 3D space. Prototypes in the Real 3D-AD training set encompass comprehensive object information from various views. The test set also includes multi-view information on objects, allowing for a more extensive exploration of the value of 3D information in AD tasks.

**3D-AD methods.** Recently, many high-quality papers have emerged in the field of 2D-AD [33, 37, 32, 26]. The release of MVTec 3D-AD also sparked interest in 3D-AD anomaly detection methods [15, 23, 5, 28, 8]. However, more research on 3D than 2D anomaly detection still needs to be done. Some methods only use depth information to remove background noise, which limits the use of depth information. Meanwhile, combining RGB and depth information without compromising performance remains a challenge. Bergmann *et al.* [1] propose a point cloud feature extraction network based on the teacher-student model. During training, the student and teacher networks maintain consistent features and use the differences in extracted features to locate anomalies during testing. Horwitz *et al.* [15] combine hand-crafted 3D descriptors with the classical AD approach KNN framework. While both of these methods are effective, their performance is poor. AST [23] performs well in MVTec 3D-AD but only uses depth information to remove background. AST still uses the 2D-AD method to detect anomalies, ignoring the depth information of the object. M3DM [28] extracts features from point clouds and RGB images separately and fuses them for better decision-making. This approach is superior to BTF but relies heavily on pre-trained large models and memory libraries. CPMF [6] also uses the KNN paradigm. However, it projects the point cloud into a two-dimensional image from different angles, significantly reducing feature extraction's complexity and computational cost and fusing the resulting information for detection. In summary, existing 3D-AD models either perform poorly or rely heavily on pre-trained models and memory libraries. Currently, there is a lack of anomaly detection methods that use point cloud information, and the available datasets for research in this field are only MVTec 3D-AD with depth information and artificially synthesized Eyescandies [3] datasets. To bring attention and research to this area, we introduce the Real3D-AD dataset.

## 3  Real3D-AD dataset

### 3.1  Data Collection

We outline the pipeline for generating the Real3D-AD dataset, including the description of a high-resolution scanner, the construction of prototypes, the generation of anomalies, and an assessment of the labor and time required for the process.

Table 1: Comparison of data collection equipment.

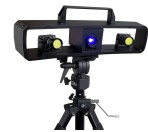

| Scanner | PMAX-S130 | Zivid One-Plus[a] |
|---|---|---|
| Dataset | Real3D-AD (Ours) | MVTec 3D-AD [1] |
| FOV | 100cm to 400cm | 60cm to 200cm |
| Point Precision | 0.011mm-0.015mm | 0.11mm |
| Spatial Distance | 0.04mm-0.07mm | 0.37mm |
| 3D Format | ASC, PLY, STL, OBJ, IGES | TIFF |

Figure 2: PMAX-S130.

[a]https://www.zivid.com

**Description of high-resolution and high-precision 3D scanner.** To obtain precise 3D anomaly detection data, we utilize a high-resolution binocular 3D scanner called PMAX-S130, as illustrated in Figure 2. The PMAX-S130 optical system comprises a pair of lenses with low distortion properties, a high luminance LED, and a blue-ray filter. The blue light scanner has a lens filter that selectively allows only the blue light of a specific wavelength to pass through. The filter effectively screens most blue light due to its relatively low concentration in natural and artificial lighting. Nevertheless, using blue light-emitting light sources could pose a unique obstacle in this context. The image sensor can collect light using the lens aperture. Hence, the influence exerted by ambient light is vastly reduced. The device exhibits the ability to perform scanning operations under intricate lighting circumstances that are frequently encountered in workshop environments. The abovementioned objective is accomplished by employing a cold light source of high-brightness LEDs. This approach prolongs the device's longevity and reduces heat emissions while ensuring consistent scanning precision. Moreover, the precision of the device's scanning is augmented by integrating a lens with minimal distortion. The data presented in Table 1 demonstrates that the PMAX-S130 performs better than the Zivid camera (utilized by MVTec 3D-AD), particularly regarding point precision. Real3D-AD exhibits a higher point precision and spatial distance per cloud than MVTec 3D-AD, with a factor of 10 and 4.28, respectively. Thus, Real3D-AD provides a pathway to enhance comprehension in high-precision point cloud anomaly detection.

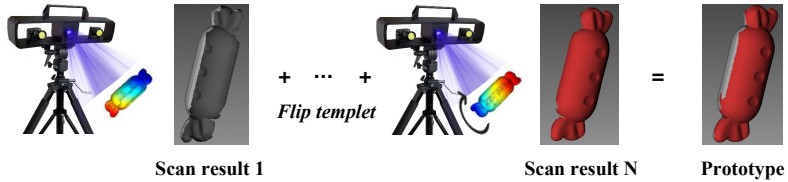

**Scan result 1**      **Scan result N**    **Prototype**

Figure 3: A prototype in the training set is made from two or more scan results.

**Prototype construction.** The prototype construction process is shown in Figure 3. Initially, the stationary object undergoes scanning while the turntable completes a full revolution of 360°, enabling the scanner to capture images of the various facets of the object. Subsequently, the object undergoes reversal, and the process of rotation and scanning is reiterated. Following the manual calibration of the front and back scanning outcomes, the algorithm precisely calibers the stitching process. If there are any gaps in the outcome, the scan stitching process is reiterated until the point cloud is rendered.

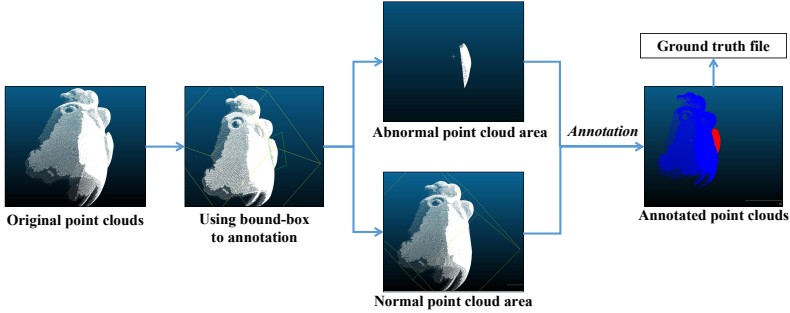

Figure 4: Anomalies annotation in Real3D-AD.

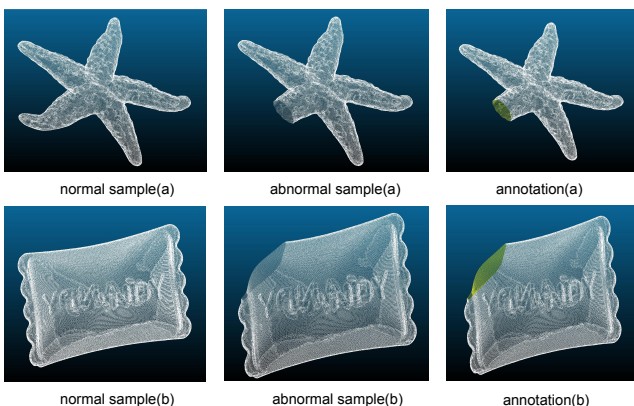

Figure 5: Incompleteness annotation in Real3D-AD. The first column refers to the normal sample. The second column denotes the abnormal sample. The third column presents the annotation for the incompleteness. Specific, we mark the cross-section and the broken edges as abnormalies, which do not introduce the extra points in the annotation.

**Anomalies types and labeling.** The anomalies pertaining to point clouds can be classified into two categories: incompleteness and redundancy. CloudCompare (2016) [10] is utilized to annotate the point cloud data. The process of labeling is depicted in Figure 4. The first step involves importing a *pcd* file into the CloudCompare software and modifying the angle view and point cloud file size. Afterward, the anomalous and non-anomalous regions were segregated and assigned corresponding labels for each point cloud. The ultimate outcome is presented in a text file.

**Labor and time-consuming.** A significant requirement for labor characterizes the process of collecting and labeling the Real3D-AD dataset. Each prototypical construction requires 1.2 days

to complete each object with a team of three individuals. The initial individual assumes the task of conducting a scan, while the subsequent individual directs their attention toward manual calibration. The third individual is primarily responsible for the labeling aspect of the task. To generate anomalies, a team of four individuals is utilized to complete the task. The initial individual directs their attention towards the inadequacy of point cloud anomalies, while the subsequent individual assumes responsibility for the superfluousness of point cloud anomalies. The second individual is primarily concerned with the process of labeling anomalies. Each atypical specimen requires 5 hours to complete. Real3D-AD involves a team of seven individuals and a time frame of four months to complete, owing to the substantial workload involved.

## 3.2 Data Statistics

The statistical information of the Real3D-AD dataset is presented in Table 2. The table consists of the dataset category, the number of training prototypes, the number of normal and abnormal samples in the test dataset, the mean proportion of abnormal points in the test. Real3D-AD comprises a total of 1,254 samples that are distributed across 12 distinct categories. Each training set for a specific category contains only four samples, similar to the few-shot scenario in 2D anomaly detection. These categories include but are not limited to Airplane, Candybar, Chicken, Diamond, Duck, Fish, Gemstone, Seahorse, Shell, Starfish, and Toffees. All these categories are toys from manufacturing lines. The data presented in Table 2 demonstrates that the low anomaly point ratio poses a challenge for detecting anomalies. The majority of the attributes in the category pertain to transparency, indicating that the Real3D-AD dataset is highly suitable for tasks involving the detection of anomalies in point clouds.

Table 2: The statistics of Real3D-AD. Note that $\Delta$ refers to the proportion of abnormal points in abnormal samples.

| | Category | Real Size [mm] | | | Attribute | Training | Test | | Total | Anomaly Point Ratio |
|---|---|---|---|---|---|---|---|---|---|---|
| | | Length | Width | Height | | Normal | Normal | Abnormal | | $\Delta$ |
| | Airplane | 34.0 | 14.2 | 31.7 | Transparency | 4 | 50 | 50 | 104 | 1.18% |
| | Car | 35.0 | 29.0 | 12.5 | Transparency | 4 | 50 | 50 | 104 | 1.99% |
| | Candybar | 33.0 | 20.0 | 8.0 | Transparency | 4 | 50 | 50 | 104 | 2.37% |
| | Chicken | 25.0 | 14.0 | 20.0 | White | 4 | 52 | 54 | 110 | 4.39% |
| | Diamond | 29.0 | 29.0 | 18.7 | Transparency | 4 | 50 | 50 | 104 | 5.41% |
| | Duck | 30.0 | 22.2 | 29.4 | Transparency | 4 | 50 | 50 | 104 | 2.00% |
| | Fish | 37.7 | 24.0 | 4.0 | Transparency | 4 | 50 | 50 | 104 | 2.86% |
| | Gemstone | 22.5 | 18.8 | 17.0 | Transparency | 4 | 50 | 50 | 104 | 2.06% |
| | Seahorse | 38.0 | 11.2 | 3.5 | Transparency | 4 | 50 | 50 | 104 | 4.57% |
| | Shell | 21.7 | 22.0 | 7.7 | Transparency | 4 | 52 | 48 | 104 | 2.25% |
| | Starfish | 27.4 | 27.4 | 4.8 | Transparency | 4 | 50 | 50 | 104 | 4.47% |
| | Toffees | 38.0 | 12.0 | 10.0 | Transparency | 4 | 50 | 50 | 104 | 2.46% |
| | Mean | 30.9 | 20.3 | 13.9 | — | 4 | 50 | 50 | 104 | 3.00% |
| | Total | — | — | — | — | 48 | 604 | 602 | 1254 | — |

Furthermore, a box-and-whisker plot represents the distribution of data points across all samples in the Real3D-AD dataset, as depicted in Figure 6. Two inferences can be made from the illustration in Figure 6. The observed differences in point count variability among distinct item categories within the point cloud demonstrate notable dissimilarity. In specific, the training samples provide complete prototypes of 3D objects while the test samples are scanned on only one side. Therefore, the number of training samples is much larger than that of test samples. Secondly, it can be observed that the disparity in point values between the normal and abnormal samples is comparatively minor within each test set.

## 3.3 Real3D-AD and other datasets

The findings presented in Table 3 demonstrate that Real3D-AD exhibits superior performance compared to MVTec 3D-AD [20] and Eyescandies [3], particularly in terms of point resolution, point precision, absence of blind spots, and dataset authenticity. Real3D-AD demonstrates a point resolution and precision of 0.04mm and 0.011mm, respectively. This is notably higher than MVTec 3D-AD, with a factor of 4.28 and 9 for point resolution and precision, respectively. Furthermore, the Real3D-AD system benefits from multi-view scanning, which eliminates any potential blind spots and thereby improves its anomaly detection capabilities. Therefore, Real3D-AD is deemed more suitable for achieving high levels of precision in point cloud anomaly detection and can accommodate industrial manufacturing requirements.

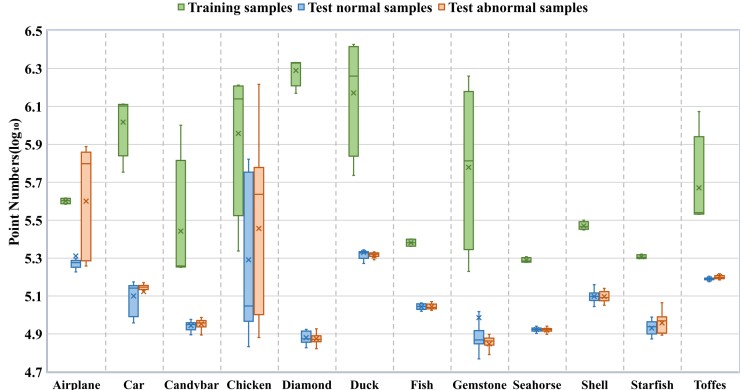

Figure 6: Point numbers for all samples on a logarithmic scale, visualized by a box-and-whisker plot.

Table 3: Comparison results of main datasets.

| Dataset | MVTec 3D-AD | Eyescandies | Real 3D-AD (Ours) |
|---|---|---|---|
| Point Resolution | 0.37mm | Not applicable | **0.04mm** |
| Point Precision | 0.11mm | Not applicable | **0.011mm** |
| All Views (No Blind Spot) | ✗ | ✗ | ✓ |
| Real/Synthesis | Real | Synthesis | Real |

## 4 Benchmark and Baseline

### 4.1 Problem Definition and Challenges

**Problem definition.** ADBENCH-3D setting can be formally stated as follows. Given a set of training examples $\mathcal{T} = \{t_i\}_{i=1}^N$, in which $\{t_1, t_2, \cdots, t_N\}$ are the training prototypes. In Real3D-AD, the number of prototypes for each category is limited($\leq 4$). In addition, $\mathcal{T}_n$ belongs to one certain category, $c_j \in \mathcal{C}$, where $\mathcal{C}$ denotes the set of all categories. During test time, given a normal or abnormal sample from a target category $c_j$, the AD model should predict whether or not the test 3D object is anomalous and localize the anomaly region if the prediction result is anomalous.

**Training and test samples visualization.** Figure 7 shows the training prototype and test dataset. The images in the blue box labeled (a)-(d) represent the prototype. The training prototype undergoes a **360°** scan, ensuring no areas of limited visibility exist. The images in the orange box (e)-(h) represent the test sample. **To simulate real-world conditions, the test sample is scanned on only one side. Since we desire to follow the real-world application: the workers or quality inspection equipments in the production line randomly check one side of the product to identify the defects by matching the scanned data with the prototype.**

**Challenges.** The following are the three challenges. (1) Each category's training dataset contains only normal prototypes, i.e., no object or point-level annotations. (2) There are few normal prototypes of the training set available. There are fewer than four training prototypes in the setting of ADBENCH-3D. (3) There are unavoidable differences between the test set and the training set samples, which need to be addressed.

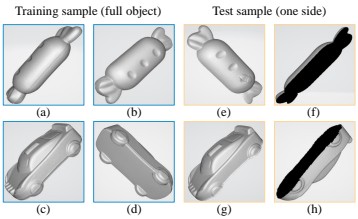

Figure 7: Examples of training and test samples in Real3D-AD.

### 4.2 ADBENCH-3D

**Metrics.** We standardize evaluation using metrics designed for 3D anomaly detection, including Area Under the Receiver Operating Characteristic Curve (AUROC) and the Area Under the

Table 4: ADBENCH-3D for Real3D-AD. The score indicates object-level AUROC ↑. The best results are highlighted in bold.

| Category | BTF | | M3DM | | PatchCore | | | Reg3D-AD |
|---|---|---|---|---|---|---|---|---|
| | Raw | FPFH | PointMAE | PointBERT | FPFH | FPFH+Raw | PointMAE | |
| Airplane | 0.730 | 0.520 | 0.434 | 0.407 | **0.882** | 0.848 | 0.726 | 0.716 |
| Car | 0.647 | 0.560 | 0.541 | 0.506 | 0.590 | **0.777** | 0.498 | 0.697 |
| Candybar | 0.539 | 0.630 | 0.552 | 0.562 | 0.541 | 0.570 | 0.663 | **0.685** |
| Chicken | 0.789 | 0.432 | 0.683 | 0.673 | 0.837 | **0.853** | 0.827 | 0.852 |
| Diamond | 0.707 | 0.545 | 0.602 | 0.627 | 0.574 | 0.784 | 0.783 | **0.900** |
| Duck | 0.691 | **0.784** | 0.433 | 0.466 | 0.546 | 0.628 | 0.489 | 0.584 |
| Fish | 0.602 | 0.549 | 0.540 | 0.556 | 0.675 | 0.837 | 0.630 | **0.915** |
| Gemstone | **0.686** | 0.648 | 0.644 | 0.617 | 0.370 | 0.359 | 0.374 | 0.417 |
| Seahorse | 0.596 | **0.779** | 0.495 | 0.494 | 0.505 | 0.767 | 0.539 | 0.762 |
| Shell | 0.396 | **0.754** | 0.694 | 0.577 | 0.589 | 0.663 | 0.501 | 0.583 |
| Starfish | 0.530 | **0.575** | 0.551 | 0.528 | 0.441 | 0.471 | 0.519 | 0.506 |
| Toffees | 0.703 | 0.462 | 0.450 | 0.442 | 0.565 | 0.626 | 0.585 | **0.827** |
| Average | 0.635 | 0.603 | 0.552 | 0.538 | 0.593 | 0.682 | 0.594 | **0.704** |

Precision-Recall curve (AUPR/AP). The details of metrics are introduced in the supplementary material.

**Methods.** As discussed in Section 2, 3D anomaly detection methods mainly focus on RGBD anomaly detection but not point anomaly detection tasks. So we adopt BTF [15] and M3DM [28] as our benchmark methods. We build a systematic benchmark, ADBENCH-3D, for our proposed Real3D-AD dataset, as shown in Table 4. In Table 4, BTF(Raw) refers to that we just adopt the coordinate features (xyz) into the BTF pipeline. BTF(FPFH) denotes we incorporate fast point feature histogram (FPFH) [24] into the BTF pipeline. M3DM(PointMAE) denotes M3DM using PointMAE [19] as a point cloud feature extractor and ignoring the RGB branch. M3DM(PointBERT) denotes M3DM using PointBert [34] as a point cloud feature extractor and ignoring the RGB branch. PatchCore+FPFH indicates we replace ResNet [14] as FPFH feature extractor and merge it into PathCore [21]. PatchCore+FPFH+Raw indicates we use FPFH and coordinates of each point cloud's feature and inject them into the PatchCore pipeline. PatchCore+PointMAE denotes that we adopt the PointMAE feature extractor and merge the feature into PatchCore architecture.

**Toolkit.** To complement ADBENCH-3D, we release a comprehensive toolkit as a starter code for high-precision point cloud anomaly detection, which implements 8 core methods in (1) data preprocessing, (2) evaluation scripts and metrics, and (3) visualization toolkit. Due to the page length limit, the toolkit details are put into the Github Repo.

## 4.3 Reg3D-AD

Inspired by PatchCore [22], we develop a general-purpose registration-based point cloud anomaly detection method (Reg3D-AD), as shown in Figure 8, which greatly meets the demands of Real3D-AD. Reg3D-AD utilizes a dual-feature representation approach to preserve the training prototypes' local and global features. Two distinct features are present in the dataset under consideration. The first feature pertains to the coordinate values of each point cloud, namely the x-, y-, and z-values. The second feature is the PointMAE feature, which characterizes the training prototypes in their entirety. The coordinate value encapsulates the localization attribute of individual points, whereas the PointMAE model prioritizes attaining a comprehensive representation of the training prototypes. The training phase aims to establish a repository of neighborhood-sensitive characteristics derived from all regular prototypes intended to serve as a memory bank. Before incorporating the novel functionalities into the memory repository, we implement the Coreset sampling technique to preserve the memory bank's size.

**Anomaly score calculation.** Before anomaly score calculation, the test 3D object needs to be registered via RANSAC algorithm [2]. After finishing the registration, the test 3D object is predicted as an anomaly if at least one point cloud is anomalous, and point-level anomaly segmentation is computed via the mean score of the point-level feature and global feature. In particular, with local feature bank $\mathcal{M}^l$ and global feature bank $\mathcal{M}^g$, the object-level anomaly scores $s$ for the test object $x^{test}$ is computed by the mean value of local feature anomaly score $s^l$ and the global feature anomaly score $s^g$. The local feature anomaly score is the maximum score $s^{l*}$ between the test 3D object's

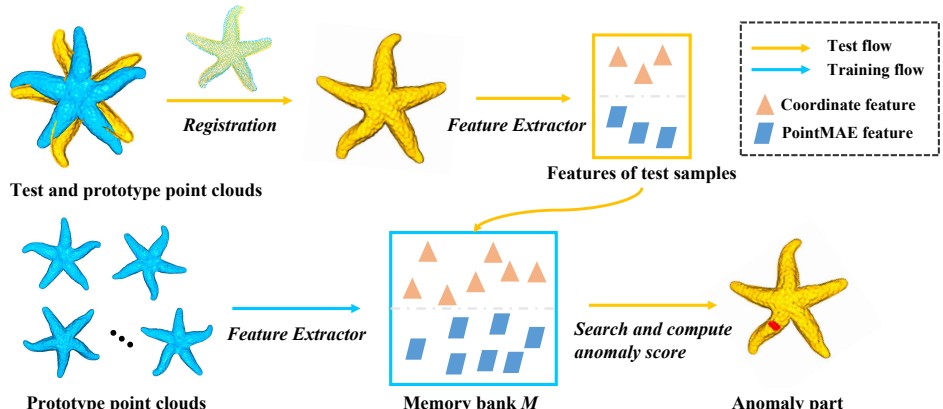

Figure 8: Pipeline of our baseline method. We extract features from the training set and sample the most representative features to the memory bank during training. During inference, we use the prototype as the target to calibrate the test sample and then extract the characteristics of the test sample to compare with the memory bank. We compute the anomaly score for each point according to the distance between test features and the memory bank.

point-level feature $\mathcal{P}(x^{test})$ and its respective nearest neighbour $m^{l*}$ in $\mathcal{M}^l$:

$$m^{test,*} = \underset{m^{test}\in\mathcal{P}(x^{test})}{\arg\max} \underset{m^l\in\mathcal{M}^l}{\min} \left\|m^{test}-m^l\right\|_2, \quad m^{l*} = \underset{m^l\in\mathcal{M}^l}{\arg\min} \left\|m^{test}-m^l\right\|_2, \quad (1)$$

$$s^{l*} = \left\|m^{test,*}-m^{l*}\right\|_2. \quad (2)$$

To enhance the robustness of the anomaly detection model, PatchCore employs an important re-weighting method [18] to tune the anomaly score:

$$s^l = \left(1 - \frac{\exp\left\|m^{test,*}-m^*\right\|_2}{\sum_{m\in\mathcal{N}_b(m^*)}\exp\left\|m^{test,*}-m\right\|_2}\right) \cdot s^{l*}, \quad (3)$$

where $\mathcal{N}_b(m^*)$ denotes $b$ nearest patch-features in $\mathcal{M}$ for test patch-feature $m^*$. The calculation of global feature anomaly scores $s^g$ is similar to $s^l$ and is achieved via the global feature memory bank $M^g$. Finally, the total anomaly score of each point cloud $s^t = (s^l + s^g)/2$.

**Analysis of ADBENCH-3D.** The findings presented in Table 4 indicate that most of the 3D anomaly detection algorithms do not meet the requirements for Real3D-AD. Upon revisiting the setting outlined in Section 4.1, there appear to be notable similarities between this setting and that of few-shot anomaly detection. This is due to the fact that the training dataset for each category is comprised of a mere 4 prototypes. The majority of contemporary state-of-the-art 3D anomaly detection algorithms are not specifically designed for tasks involving the detection of anomalies in a few-shot scenario. To tackle this challenge, it is imperative to optimize the utilization of prototype data and guarantee that the acquired point cloud data remains unaffected by spatial relative positions. From Table 4, it can be clearly shown that our baseline method, Reg3D-AD, outperforms state-of-the-art 3D anomaly detection methods for the Real3D-AD dataset.

## 5   Limitations & Potential negative societal impacts

**Limitations.** There is still a broad scope for improvement and exploration based on our work. For example, our data is sourced from the 3D scanner and only contains spatial information, a common practice in industrial production. However, obtaining standardized RGB point cloud templates may be possible by calibrating and stitching multiple RGBD images or using modeling software rendering. RGB point cloud templates may be simultaneously applied to RGB image (2D) anomaly detection and point cloud (3D) anomaly detection. Additionally, our data can generate depth images from different angles by controlling rendering conditions, enabling anomaly detection from that perspective. This

has yet to be explored. Furthermore, although our baseline outperforms existing anomaly detection methods, it is still susceptible to false detection because the test point cloud edges are truncated. Therefore, more advanced models are expected to address these issues more effectively. Our work, as a first attempt at full-view point cloud anomaly detection, will inspire further exploration in this field.

**Potential negative societal impacts.** Our data is obtained from scanning industrial products, so no negative social impact will exist.

## 6  Conclusion

In this work, we propose a Real3D-AD dataset to investigate high-precision point cloud anomaly detection problems, which aims to facilitate the research of defect identification for advancing machining and precision manufacturing. The most extraordinary high-precision 3D industrial anomaly detection dataset to date, Real3D-AD, comprises 1,254 high-resolution 3D items ($\geq$ one million point clouds for each item) spanning 12 objects of real-world applicability. Regarding point cloud resolution (0.0010mm-0.0015mm), $360°$ degree coverage, and flawless prototype, Real3D-AD outperforms currently available 3D anomaly detection datasets. In addition, we provide a thorough assessment of Real3D-AD datasets, highlighting the absence of baseline approaches to enable high-precision point cloud anomaly detection applications. We put forth a general registration-based 3D anomaly detection technique (Reg3D-AD) and 3D feature coupling unit that keeps local features and global representations. Experiments on the Real3D-AD dataset show that Reg3D-AD performs significantly better than the next-best approach.

**Acknowledgments**. This work is supported by the National Key R&D Program of China (Grant NO. 2022YFF1202903) and the National Natural Science Foundation of China (Grant NO. 62122035, 62206122).

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

## A  Experiment setup

Due to the limitation of GPU memory, we make subsampling on benchmark methods. For BTF [15], we sample and store the points of training samples at a ratio of 100:1, and for testing samples, we sample the points at a ratio of 500:1. After calculating the anomaly scores for each point, we estimate the anomaly scores for the unsampled points using the k-nearest neighbors (KNN) algorithm [13]. As for M3DM [28], we set the point groups number of point transformer to 16384. In our experiments based on PatchCore [22], we uniformly set the size of the memory bank to 10000.

## B  Experiments

Due to the page limit, we only give the object level auroc results in the main text, here we show the object level au-pr in Table 5, point level auroc in Table 6 and point level au-pr in Table 7. Among all the methods, our Reg3D-AD only slightly underperforms BTF (Raw) in terms of point-level AUROC, and slightly underperforms our self-designed PatchCore (FPFH+Raw) in terms of point-level AUPR. Overall, Reg3D-AD remains a reliable baseline. In addition, we adjusted different memory bank sizes and PointMAE point group numbers for comparison. As shown in Table 8, a larger PointMAE point group number improves model performance, while a larger memory bank does not always lead to performance improvements.

Table 5: ADBENCH-3D for Real3D-AD. The score indicates object-level AUPR ↑.

| Category | BTF | | M3DM | | PatchCore | | | Reg3D-AD |
|---|---|---|---|---|---|---|---|---|
| | *Raw* | *FPFH* | *PointMAE* | *PointBERT* | *FPFH* | *FPFH+Raw* | *PointMAE* | |
| airplane | 0.659 | 0.506 | 0.479 | 0.497 | 0.852 | 0.807 | 0.747 | 0.703 |
| car | 0.653 | 0.523 | 0.508 | 0.517 | 0.611 | 0.766 | 0.555 | 0.753 |
| candybar | 0.638 | 0.490 | 0.498 | 0.480 | 0.553 | 0.611 | 0.576 | 0.824 |
| chicken | 0.814 | 0.464 | 0.739 | 0.716 | 0.872 | 0.885 | 0.864 | 0.884 |
| diamond | 0.677 | 0.535 | 0.620 | 0.661 | 0.569 | 0.767 | 0.801 | 0.884 |
| duck | 0.620 | 0.760 | 0.533 | 0.569 | 0.506 | 0.560 | 0.488 | 0.588 |
| fish | 0.638 | 0.633 | 0.525 | 0.628 | 0.642 | 0.844 | 0.720 | 0.939 |
| gemstone | 0.603 | 0.598 | 0.663 | 0.628 | 0.411 | 0.411 | 0.444 | 0.454 |
| seahorse | 0.567 | 0.793 | 0.518 | 0.491 | 0.508 | 0.763 | 0.546 | 0.787 |
| shell | 0.434 | 0.751 | 0.616 | 0.638 | 0.573 | 0.553 | 0.590 | 0.646 |
| starfish | 0.557 | 0.579 | 0.573 | 0.573 | 0.491 | 0.473 | 0.561 | 0.491 |
| toffees | 0.505 | 0.700 | 0.593 | 0.569 | 0.506 | 0.559 | 0.708 | 0.721 |
| Average | 0.624 | 0.603 | 0.572 | 0.581 | 0.599 | 0.676 | 0.626 | 0.723 |

Table 6: ADBENCH-3D for Real3D-AD. The score indicates point-level AUROC ↑.

| Category | BTF | | M3DM | | PatchCore | | | Reg3D-AD |
|---|---|---|---|---|---|---|---|---|
| | *Raw* | *FPFH* | *PointMAE* | *PointBERT* | *FPFH* | *FPFH+Raw* | *PointMAE* | |
| airplane | 0.738 | 0.564 | 0.530 | 0.523 | 0.471 | 0.556 | 0.579 | 0.631 |
| car | 0.708 | 0.647 | 0.607 | 0.593 | 0.643 | 0.740 | 0.610 | 0.718 |
| candybar | 0.864 | 0.735 | 0.683 | 0.682 | 0.637 | 0.749 | 0.635 | 0.724 |
| chicken | 0.693 | 0.608 | 0.735 | 0.790 | 0.618 | 0.558 | 0.683 | 0.676 |
| diamond | 0.882 | 0.563 | 0.618 | 0.594 | 0.760 | 0.854 | 0.776 | 0.835 |
| duck | 0.875 | 0.601 | 0.678 | 0.668 | 0.430 | 0.658 | 0.439 | 0.503 |
| fish | 0.709 | 0.514 | 0.600 | 0.589 | 0.464 | 0.781 | 0.714 | 0.826 |
| gemstone | 0.891 | 0.597 | 0.654 | 0.646 | 0.830 | 0.539 | 0.514 | 0.545 |
| seahorse | 0.512 | 0.520 | 0.561 | 0.574 | 0.544 | 0.808 | 0.660 | 0.817 |
| shell | 0.571 | 0.489 | 0.748 | 0.732 | 0.596 | 0.753 | 0.725 | 0.811 |
| starfish | 0.501 | 0.392 | 0.555 | 0.563 | 0.522 | 0.613 | 0.641 | 0.617 |
| toffees | 0.815 | 0.623 | 0.679 | 0.677 | 0.411 | 0.549 | 0.727 | 0.759 |
| Average | 0.722 | 0.566 | 0.637 | 0.636 | 0.592 | 0.692 | 0.634 | 0.700 |

Table 7: ADBENCH-3D for Real3D-AD. The score indicates point-level AUPR ↑.

| Category | BTF | | M3DM | | PatchCore | | | Reg3D-AD |
|---|---|---|---|---|---|---|---|---|
| | *Raw* | *FPFH* | *PointMAE* | *PointBERT* | *FPFH* | *FPFH+Raw* | *PointMAE* | |
| airplane | 0.027 | 0.012 | 0.007 | 0.007 | 0.027 | 0.016 | 0.016 | 0.017 |
| car | 0.028 | 0.014 | 0.018 | 0.017 | 0.034 | 0.160 | 0.069 | 0.135 |
| candybar | 0.118 | 0.025 | 0.016 | 0.016 | 0.142 | 0.092 | 0.020 | 0.109 |
| chicken | 0.044 | 0.049 | 0.310 | 0.377 | 0.040 | 0.045 | 0.052 | 0.044 |
| diamond | 0.239 | 0.032 | 0.033 | 0.038 | 0.273 | 0.363 | 0.107 | 0.191 |
| duck | 0.068 | 0.020 | 0.011 | 0.011 | 0.055 | 0.034 | 0.008 | 0.010 |
| fish | 0.036 | 0.017 | 0.025 | 0.039 | 0.052 | 0.266 | 0.201 | 0.437 |
| gemstone | 0.075 | 0.014 | 0.018 | 0.017 | 0.093 | 0.066 | 0.008 | 0.016 |
| seahorse | 0.027 | 0.031 | 0.030 | 0.028 | 0.031 | 0.291 | 0.071 | 0.182 |
| shell | 0.018 | 0.011 | 0.022 | 0.021 | 0.031 | 0.049 | 0.043 | 0.065 |
| starfish | 0.034 | 0.017 | 0.040 | 0.040 | 0.037 | 0.035 | 0.046 | 0.039 |
| toffees | 0.055 | 0.016 | 0.021 | 0.018 | 0.040 | 0.055 | 0.055 | 0.067 |
| Average | 0.065 | 0.022 | 0.046 | 0.052 | 0.074 | 0.129 | 0.058 | 0.113 |

Table 8: Ablation experiment for our Reg3D-AD. We modified the point groups number of Point Transformer and the memory size of the memory bank. Note that O in O-AUROC refers to object-level, and P in P-AUROC refers to point-level.

| Category | Ours(point groups 16384, memory size 10000) | | | | Ours(point groups 8192, memory size 10000) | | | | Ours(point groups 16384, memory size 20000) | | | |
|---|---|---|---|---|---|---|---|---|---|---|---|---|
| | O-AUROC | P-AUROC | O-AUPR | P-AUPR | O-AUROC | P-AUROC | O-AUPR | P-AUPR | O-AUROC | P-AUROC | O-AUPR | P-AUPR |
| airplane | 0.716 | 0.631 | 0.703 | 0.017 | 0.873 | 0.704 | 0.819 | 0.024 | 0.737 | 0.622 | 0.761 | 0.014 |
| car | 0.697 | 0.718 | 0.753 | 0.135 | 0.752 | 0.777 | 0.744 | 0.183 | 0.719 | 0.735 | 0.706 | 0.170 |
| candybar | 0.827 | 0.724 | 0.824 | 0.109 | 0.693 | 0.751 | 0.709 | 0.077 | 0.732 | 0.704 | 0.744 | 0.070 |
| chicken | 0.852 | 0.676 | 0.884 | 0.044 | 0.621 | 0.457 | 0.628 | 0.040 | 0.648 | 0.689 | 0.636 | 0.052 |
| diamond | 0.900 | 0.835 | 0.884 | 0.191 | 0.794 | 0.792 | 0.794 | 0.297 | 0.882 | 0.817 | 0.877 | 0.122 |
| duck | 0.584 | 0.503 | 0.588 | 0.010 | 0.570 | 0.342 | 0.515 | 0.012 | 0.572 | 0.521 | 0.564 | 0.012 |
| fish | 0.915 | 0.826 | 0.939 | 0.437 | 0.836 | 0.818 | 0.848 | 0.325 | 0.948 | 0.825 | 0.953 | 0.418 |
| gemstone | 0.417 | 0.545 | 0.454 | 0.016 | 0.386 | 0.737 | 0.421 | 0.087 | 0.420 | 0.544 | 0.458 | 0.032 |
| seahorse | 0.762 | 0.817 | 0.787 | 0.182 | 0.713 | 0.773 | 0.702 | 0.191 | 0.726 | 0.811 | 0.707 | 0.202 |
| shell | 0.583 | 0.811 | 0.646 | 0.065 | 0.665 | 0.753 | 0.574 | 0.042 | 0.542 | 0.803 | 0.583 | 0.061 |
| starfish | 0.506 | 0.617 | 0.491 | 0.039 | 0.422 | 0.591 | 0.432 | 0.039 | 0.485 | 0.593 | 0.460 | 0.034 |
| toffees | 0.685 | 0.759 | 0.721 | 0.067 | 0.562 | 0.419 | 0.554 | 0.036 | 0.710 | 0.764 | 0.744 | 0.071 |
| Average | 0.705 | 0.700 | 0.723 | 0.113 | 0.666 | 0.681 | 0.653 | 0.120 | 0.674 | 0.697 | 0.677 | 0.108 |

