# OpenReview forum: "Real3D-AD: A Dataset of Point Cloud Anomaly Detection"
_NeurIPS.cc/2023/Track/Datasets_and_Benchmarks — NeurIPS 2023 Datasets and Benchmarks Poster_

### Official Review · Reviewer_KAeA · 2023-07-05
**Real3D-AD makes a huge impact on the field of 3D anomaly detection and fills the gap between industrial manufacturing and the academy.**

**Rating:** 9
**Confidence:** 4

**Strengths:**

+ The paper presents a novel promising scene for 3D anomaly detection. The paper posits that the detection of anomalies in industrial products can be achieved through the training of a model with a limited number of design prototypes. This is due to the fact that industrial products typically possess such prototypes. The aforementioned methodology exhibits the possibility of emerging as a novel paradigm in the realm of 3D anomaly detection.


+ The study presents a dataset for detecting anomalies in point clouds, which is tailored for the context of utilizing product prototypes for 3D anomaly detection. Compared with existing 3D anomaly detection dataset, Real3D-AD exhibits high-precision, high resolution, the absence of blind spots and broad extensive usage.


+ The authors propose a novel baseline method and suggest potential directions for future improvements, offering valuable insights and improvement strategies for subsequent researchers.


**Additional Feedback:**

PointMAE only computes features for a subset of point cloud clusters during feature extraction, hence the benchmark restricts the final number of features. Personally, I believe that downsampling the original samples using voxelization can also address this issue, thereby improving the detection accuracy at the Object level.
The Real3D-AD dataset is of great value, but current methods are still not applicable in real-world scenarios. It is hoped that the authors will continue to explore and, one day, achieve the goal of using AI to drive productivity development.

**Clarity:**

The structure and organization of the paper are reasonable, and the flow of the content is smooth. No obvious spelling errors were found.


**Correctness:**

The relevant statements in the paper are appropriate. The construction of the dataset is reasonable, and the sources of the data are provided. Additionally, the code for the benchmark has been made openly available. The benchmark evaluation method is suitable and can be easily reproduced.




**Documentation:**

The paper provides a detailed description of the dataset creation process, and there are no ethical concerns regarding its production. The authors have made the dataset openly available on GitHub under the CC BY 4.0 license. The benchmark's environment setup, dataset structure partitioning, and reproducibility methods are also thoroughly described on GitHub.

**Ethics:**

I believe that the submitted content does not raise any ethical concerns

**Limitations:**

In the supplementary materials, the authors thoroughly discuss the limitations of their work and potential directions for improvement from both the data and method perspectives. The dataset created in the paper is purely based on industrial product and does not have any negative implications.


**Opportunities For Improvement:**

+ The descriptions in the paper need to be checked again carefully, such as the statement in Section 4: 'The images in the blue box labeled (a)-(d) represent the prototype,' where (c)-(d) do not seem to be prototypes.

+ From Figure 7, it can be observed that the training samples are not aligned. However, the absence of alignment among the training samples may have an impact on the model's performance. It is unclear from the paper whether this aspect was omitted.

+ It is not explicitly stated how PatchCore (FPFH+Raw) combines FPFH and Raw features. Additionally, it is unclear from the paper whether Reg3D-AD normalizes the retrieval scores from the two memory banks before integrating them.

**Relation To Prior Work:**

This paper introduces a point cloud anomaly detection dataset in a novel scenario, which results in a relatively low overlap with previous works. The paper provides a detailed discussion in the “Introduction” about the differences between this dataset and previous RGBD anomaly detection datasets in Table 3.

**Summary And Contributions:**

Real3D-AD offers a point cloud dataset that facilitates realistic 3D anomaly detection, with the objective of catering to the requirements of industrial manufacturing. This paper provides a comprehensive account of the Real3D-AD generation procedure, which involves the utilization of a high-precision and high-resolution blue-ray scanner for capturing the requisite details. Furthermore, this manuscript presents the annotation methods employed for Real3D-AD, which necessitates a significant expenditure of labor. Furthermore, the present study provides a clear demonstration of the benefits associated with Real3D-AD, which contains superior point resolution, heightened point precision, absence of blind spots, and the utilization of real-world objects in data capture. This paper presents a novel approach for simulating real-world 3D anomaly detection and establishes a comprehensive benchmark using this approach. Ultimately, the authors present an innovative foundational approach for Real3D-AD.


The main contributions of the paper can be summarized as follows:

+ This paper proposes a high-precision and superior resolution point cloud dataset to adapt the demands of 3D anomaly detection. Moreover, the paper gives the detail of the data collection, processing, annotation and the statistics, which clearly illustrate the whole picture of Real3D-AD


+ This paper introduces an application-specific setting, which caters the requirement of real-world 3D anomaly detection. Based on this setting, the authors provide a systematic benchmark and the starter code of data preprocessing, evaluation metrics and scripts.


+ This manuscript offers a generalized 3D anomaly detection approach for Real3D-AD. The extensive experiments proves  the high efficacy and performance.

Overall, the paper's main contributions lie in introducing a novel 3D anomaly detection setting, providing a high-quality point cloud dataset to simulate the scenario, and establishing a comprehensive benchmark that evaluates various methods using multiple metrics.

---

> ### Author Response · Authors · 2023-08-13
> **Response to reviewer KAeA**
>
> We sincerely thank you for your detailed comments and constructive suggestions, especially your appreciation for our work, **"a comprehensive account"**, **"a clear demonstration"**, **"a novel approach"**, **"a comprehensive benchmark"**, **"high-precision"**, **"superior-resolution"**, **"a systematic benchmark"**, **"high quality"**. Next, we respond to your concerns as follows.
>
> ***Q1: The descriptions in the paper need to be checked again carefully, such as the statement in Section 4: 'The images in the blue box labeled (a)-(d) represent the prototype,' where (c)-(d) do not seem to be prototypes.***
>
> Thanks for the reviewer's correction. We have revised it in the updated manuscript.
>
> ***Q2: From Figure 7, it can be observed that the training samples are not aligned. However, the absence of alignment among the training samples may have an impact on the model's performance. It is unclear from the paper whether this aspect was omitted.***
>
> Thanks for the reviewer's suggestion. Sorry for making you confused. It's clear that the absence of alignment certainly impacts the performance of Reg3D-AD. In the training process, we randomly select one of the training samples as the prototype, and then align the remaining training samples with the prototypes using registration algorithms.
>
> ***Q3: It is not explicitly stated how PatchCore (FPFH+Raw) combines FPFH and Raw features. Additionally, it is unclear from the paper whether Reg3D-AD normalizes the retrieval scores from the two memory banks before integrating them.***
>
> Thank you for pointing this out. We do not actually combine FPFH and Raw features. Instead, we employe them for feature retrieval, calculated abnormal scores, and then make judgments based on the results from both features. In Reg3D-AD, the normalization of the two scores takes place before combining the scores obtained from the two memory banks. For further specifics, please refer to our code. If you have any additional questions or concerns, please don't hesitate to reach out to us.

---

> ### Author Response · Authors · 2023-08-17
> **Further Discussion**
>
> Dear Reviewer KAeA:
>
> Thank you for taking the time to review our work. We hope our rebuttal has addressed your questions and concerns. We would be more than happy to discuss with you if you still have any unresolved concerns or additional questions about the paper or our rebuttal.
>
> Best,
>
> The authors of Real3D-AD

---

> > ### Comment · Reviewer_KAeA · 2023-08-21
> >
> > I thank the authors for addressing my concerns. The responses have addressed my concerns. Thus, I decide to keep my initial high score.

---

> > > ### Author Response · Authors · 2023-08-21
> > > **Thasnks Reviewer KAeA for your support for our work**
> > >
> > > Dear Reviewer KAeA,
> > >
> > > We are delighted that we were able to address your concerns. Thank you very much for your strong support for our paper. We hope that our work can offer new insights for industrial anomaly detection.

---

### Official Review · Reviewer_6MP5 · 2023-07-18
**A new dataset for 3D point cloud anomaly detection with some problems**

**Rating:** 6
**Confidence:** 4

**Strengths:**

As the number of available 3D anomaly detection datasets is very limited, a new dataset for this task is a valuable contribution to the field.

The introduced dataset is the first 3D anomaly detection dataset in which the point clouds are not just created from a single view. This introduces a novel challenge. The dataset appears to have been created with great attention to detail in order to produce a high quality of the generated point clouds.

While not the main contribution of the paper, the introduced method is a sound approach. The significant difference in performance compared to the other evaluated methods underlines this.

**Additional Feedback:**

I have one question that I hope the authors could answer: How are missing parts of the objects labelled in the point clouds? As the 3D scanner does not provide any points in those regions, it is not clear what one could annotate. Introducing new points labelled as abnormal is problematic in my opintion, as there are no defined coordinates for missing points.

I found a few typos, e.g., in line 97 "The abnormal images contain 1,148 and ...". I think a word is missing after "1,148". It would be great if the authors could check for typos again before submitting the final version of the paper.

**Clarity:**

The writing is generally fine, but I did find some issues.

Some information is not where I would expect it to be. For example, section 3 ("Real3D-AD dataset") does not contain the information that the test samples are scanned from only one view. I would expect this key information about the dataset to be in the section that introduces the dataset.

Some parts the paper are a bit confusing for me. For example, in line 181, the authors talk about the "mean proportion of abnormal point clouds". From this sentence alone, I would think that this proportion is 0.5, as there are (roughly) 50 normal and 50 abnormal points clouds per object category. However, I think it refers to the "Anomaly Ratio" in Table 2, which I assume is the mean ratio of points labelled as "abnormal" in a point cloud. In line 182, the authors mention that Table 2 shows "the minimum, maximum, and the average number of point clouds in both the training and test datasets". I do not know which part of the table this refers to. At several points in the paper, the word "point cloud" is used where I think "points" is meant. For example, in the abstract the authors mention "forty thousand to millions point clouds for each item". Should this be "forty thousand to million points per cloud"?

**Correctness:**

I see a potential problem for the evaluation metrics caused by the fact that the test samples are scanned just from a single view (as mentioned in "Opportunities for Improvement").

I think formula (1) is incorrect. Should the argmin over $m^l$ be a min? In the definition of $s^{l*}$, it is unclear what $m^{test}$ is.

**Documentation:**

The process of dataset collection is described in detail. The information in the paper and the supplementary PDF document alone does not suffice to reproduce the results of the benchmark, but the provided code should make a reproduction of the results possible.

**Ethics:**

I do not see any ethical concerns that would warrant an ethics review.

**Limitations:**

The authors discuss some limitations of the dataset and the proposed method in the supplementary material. The limitations I see (cf. "Opportunities for Improvement") are not fully addressed.

**Opportunities For Improvement:**

All object classes in the dataset appear to have a rigid form that does not change between individual exemplars of a single object class. This reduces the challenge as anomalies can basically only consist in missing or additional parts compared to the training prototypes. Objects with variations would introduce greater difficulties since a model trained on these would need to learn what normal variations are.

The training set for each object contains only 4 samples. This restriction makes approaches that require a lot of training data infeasible. In their discussion of the benchmark, the authors note that it is similar to a few-shot problem. I think this is a very important point that should be made more explicit in the introduction of the dataset. This also presents an issue for the interpretation of the relatively bad performance of the evaluated methods on this dataset. It is not clear whether this is due to an inherent difficulty of the objects in the dataset or just caused by the lack of sufficient training data.

The test samples are only scanned from one point of view, so the generated point clouds do not represent the whole object as the training samples do. This introduces a data shift that might present a problem to some methods. It might also cause problems for the evaluation metrics, in particular the point-level metrics: At the border of a point cloud, it might not be clear whether something is missing or whether it just was not captured by the 3D scanner. I am not fully convinced by the reasoning the authors provide for the single-view test samples ("Since we desire to follow the real-world application: the workers in the production line randomly check one side of the product to identify the defects by matching the scanned data with the prototype."). In their introduction of the dataset, the authors stress the absence of blind spots as an advantage compared to other datasets. As that is only true for the four training samples per object class, this advantage is somewhat limited.

**Relation To Prior Work:**

The difference to related work is sufficienty discussed.

**Summary And Contributions:**

The authors introduce a new dataset for anomaly detection in 3D point clouds. It consist of 12 different object classes. For each object class, there are 4 samples as a training set and about 50 normal and 50 anomalous samples for testing. The dataset was created using a high-precision 3D scanner on real objects, leading to very high-resolution point clouds. The training point clouds represent the object from all views (as opposed to, e.g. point clouds created from an RGBD image which covers only a single view). The test point clouds only show the object from one (random) viewpoint.

Along with the dataset, the authors also provide a full evaluation toolkit that can be used to evaluate anomaly detection methods on the dataset.

The authors also introduce a novel method for 3D point cloud anomaly detection. It is a registration-based method that compares features extracted from a point cloud to a memory bank of features generated from the training data.

Evaluations on the dataset show that the new approach outperforms the other evaluated methods, but that there is still much room for improvement for future approaches.

---

> ### Author Response · Authors · 2023-08-13
> **Response to reviewer 6MP5(Part 1)**
>
> We sincerely thank you for the detailed comments and positive feedback such as, **"valuable contribution"**, **"introduces a novel challenge"**, **"very high resolution point clouds"**, **"a full evaluation toolkit"**, **"The process of dataset collection is in detail"**, **"The difference to related work is sufficiently discussed"**, **"a sound approach"**. For each detailed question, we provide responses below.
>
> ***Q1: All object classes in the dataset appear to have a rigid form that does not change between individual exemplars of a single object class. This reduces the challenge as anomalies can basically only consist of missing or additional parts compared to the training prototypes. Objects with variations would introduce greater difficulties since a model trained on these would need to learn what normal variations are.***
>
> Thank the reviewer for his/her valuable suggestion and we kindly disagree that only two anomaly types reduce the challenge. Conversely, we maintain that Real3D-AD presents a considerable level of difficulty, which is essential for effectively evaluating the performance of 3D-AD methods. Firstly, as illustrated in Table 1, we have compiled a statistical table based on the widely recognized MVTec 3D-AD dataset. Our analysis indicates that the anomaly types can be categorized as either incompleteness or redundancy. As a result, the anomaly types proposed within Real3D-AD are representative of the intended anomalies. Moreover, the ADBench-3D results in Table 4 of the manuscript clearly demonstrate that a significant proportion of 3D-AD methods do not exhibit strong performance. This presents a substantial opportunity for researchers to devise algorithms aimed at addressing this challenge. Additionally, it's worth noting that a substantial portion of products available in the market are rigid in form. We recognise that incorporating additional variations into Real3D-AD has the potential to bring about greater diversity in anomalies and heightened difficulty. This presents a promising avenue for future investigation.
>
> ***Table 1: The statistics of anomaly types in MVTec 3D-AD[1].***
>
> |   anomaly type\object type                             | bagel         | cable_gland | carrot        | cookie        | dowel         | foam          | peach         | potato        | rope          | tire          |
> |--------------------------------|---------------|-------------|---------------|---------------|---------------|---------------|---------------|---------------|---------------|---------------|
> | incompleteness                 | crack         | bent        | crack         | crack         | bent          | cut           | cut           | cut           | cut           | cut           |
> | incompleteness                 | hole          | cut         | cut           | hole          | cut           |               | hole          | hole          | open          | hole          |
> | incompleteness                 |               | hole        | hole          |               |               |               |               |               |               |               |
> | incompleteness                 |               | thread      |               |               |               |               |               |               |               |               |
> | reduncdancy                    | contamination |             | contamination | contamination | contamination | contamination | contamination | contamination | contamination | contamination |
> | incompleteness and reduncdancy | combined      |             | combined      | combined      | combined      | combined      | combined      | combined      | combined      | combined      |
> | not included                   |               |             |               |               |               | color         |               |               |               |               |
> |

---

> > ### Author Response · Authors · 2023-08-13
> > **Response to reviewer 6MP5(Part 2)**
> >
> > ***Q2: The training set for each object contains only 4 samples. This restriction makes approaches that require a lot of training data infeasible. In their discussion of the benchmark, the authors note that it is similar to a few-shot problem. I think this is a very important point that should be made more explicit in the introduction of the dataset. This also presents an issue for the interpretation of the relatively bad performance of the evaluated methods on this dataset. It is not clear whether this is due to an inherent difficulty of the objects in the dataset or just caused by the lack of sufficient training data.***
> >
> > Thanks for the reviewer's insightful suggestion. There are several reasons for why Real3D-AD can be viewed as a few-shot problem leading to poor performance. Firstly, few-shot anomaly detection (AD) [1][2][3] and even zero-shot AD [4] have demonstrated highly competitive performance in MVTec AD [5]. This underscores the growing interest in the field of few-shot AD. However, considering the scarcity of 3D-AD methods focusing on few-shot 3D-AD and point cloud-based anomaly detection, the poor performance observed in ADBench-3D (refer to Table 4 in the manuscript) can be anticipated. Additionally, we have conducted three experiments to verify that the insufficient training dataset is not the primary cause of the poor performance in ADBench-3D. Tables 2 to 4 provide a clear demonstration that increasing the training prototypes fails to enhance performance and even leads to degradation. This solidifies our assertion that the relatively subpar performance is rooted in the inherent difficulty of the task. Lastly, as per the reviewers' suggestion, we have included a more detailed explanation within the paper, ***"Section 4.1 provides a comprehensive description of the setting. Be more specific, the training examples are limited ($\leq$ 4) and the test sample are only scanned by one side. The motivation is to simulate the real-world application: The scanning positions on the production line are fixed, and one position can only scan the results of one side of the product."***
> >
> > ***Table 2: The 3D anomaly detection performance when using 4 prototypes for training.***
> >
> >
> > | Methods | BTF_FPFH | BTF_Raw | M3DM_PointMAE | M3DM_PointBERT | PatchCore+FPFH | PatchCore+FPFH+raw | PatchCore+PointMAE | Our baseline |
> > |--------------|---------|----------|---------------|----------------|----------------|--------------------|--------------------|--------------|
> > | Object AUROC      | 0.635 |	0.603 |	0.552 |	0.538 |	0.593 |	0.682 |	0.594 |	0.704 |
> > | Object AUPR     | 0.614 |	0.611 |	0.572 |	0.581 |	0.591 |	0.667 |	0.633 |	0.723        |
> > | Point AUROC     | 0.730 |	0.571 |	0.637 |	0.636 |	0.577 |	0.680 |	0.642 |	0.705         |
> > | Point AUPR    | 0.064 |	0.022 |	0.046 |	0.052 |	0.071 |	0.123 |	0.058 |	0.109         |
> > |
> >
> > ***Table 3: The 3D anomaly detection performance when using 8 prototypes for training.***
> >
> > | Methods | BTF_FPFH | BTF_Raw     | M3DM_PointMAE | M3DM_PointBERT | PatchCore+FPFH | PatchCore+FPFH+raw | PatchCore+PointMAE | Our baseline |
> > | ------------ | -------- | ----------- | ------------- | -------------- | -------------- | ------------------ | ------------------ | ------------ |
> > | Object AUROC      | 0.615  | 0.550 | 0.525   | 0.505    | 0.583    | 0.657        | 0.585       | 0.678  |
> > | Object AUPR    | 0.596 | 0.579 | 0.539   | 0.549        | 0.569    | 0.649        | 0.616        | 0.690      |
> > | Point AUROC     | 0.734  | 0.553   | 0.628        | 0.613         | 0.570    | 0.647            | 0.631        | 0.685  |
> > | Point AUPR    | 0.067 | 0.020 | 0.023         | 0.024        | 0.066    | 0.095        | 0.058        | 0.097      |
> > |
> >
> > ***Table 4: The 3D anomaly detection performance when using 16 prototypes for training.***
> >
> > | Methods | BTF_FPFH    | BTF_Raw     | M3DM_PointMAE | M3DM_PointBERT | PatchCore+FPFH | PatchCore+FPFH+raw | PatchCore+PointMAE | Our baseline |
> > | ------------ | ----------- | ----------- | ------------- | -------------- | -------------- | ------------------ | ------------------ | ------------ |
> > | Object AUROC      | 0.597 | 0.527 | 0.525   | 0.514        | 0.581    | 0.654        | 0.585              | 0.688        |
> > | Object AUPR     | 0.597 | 0.550 | 0.546   | 0.554    | 0.566    | 0.643        | 0.617        | 0.702  |
> > | Point AUROC    | 0.737 | 0.542 | 0.628   | 0.612        | 0.516          | 0.630        | 0.627        | 0.652  |
> > | Point AUPR    | 0.061 | 0.020 | 0.023       | 0.024          | 0.057    | 0.092        | 0.055        | 0.084        |
> > |
> >
> > Due to the extensive content of our experiments, we have provided detailed data regarding different types of items on the dataset's main webpage: https://github.com/M-3LAB/Real3D-AD.

---

> > > ### Author Response · Authors · 2023-08-13
> > > **Response to reviewer 6MP5(Part 3)**
> > >
> > > ### References for question 2
> > >
> > > [1] Xie, G., Wang, J., Liu, J., Jin, Y., & Zheng, F. (2022, September). Pushing the Limits of Fewshot Anomaly Detection in Industry Vision: Graphcore. In The Eleventh International Conference on Learning Representations.
> > >
> > > [2] Huang, C., Guan, H., Jiang, A., Zhang, Y., Spratling, M., & Wang, Y. F. (2022, October). Registration based few-shot anomaly detection. In European Conference on Computer Vision (pp. 303-319). Cham: Springer Nature Switzerland.
> > >
> > > [3] Santos, J., Tran, T., & Rippel, O. (2023). Optimizing PatchCore for Few/many-shot Anomaly Detection. arXiv preprint arXiv:2307.10792.
> > >
> > > [4] Jeong, J., Zou, Y., Kim, T., Zhang, D., Ravichandran, A., & Dabeer, O. (2023). Winclip: Zero-/few-shot anomaly classification and segmentation. In Proceedings of the IEEE/CVF Conference on Computer Vision and Pattern Recognition (pp. 19606-19616).
> > >
> > > [5] Bergmann, P., Fauser, M., Sattlegger, D., & Steger, C. (2019). MVTec AD--A comprehensive real-world dataset for unsupervised anomaly detection. In Proceedings of the IEEE/CVF conference on computer vision and pattern recognition (pp. 9592-9600).
> > >
> > > ***Q3: The test samples are only scanned from one point of view, so the generated point clouds do not represent the whole object as the training samples do. This introduces a data shift that might present a problem to some methods.***
> > >
> > > Thanks for the reviewer pointing this out. Your analysis is correct. It may introduce a data shift problem for the existing 3D-AD methods. Hence, this point of view verifies that Real3D-AD is difficult enough to verify the performance since data-shift problem often occurs in real-world applications
> > >
> > > ***Q4: It might also cause problems for the evaluation metrics, in particular the point-level metrics: At the border of a point cloud, it might not be clear whether something is missing or whether it just was not captured by the 3D scanner.***
> > >
> > > Thanks for the reviewer pointing this out. We acknowledge that our explanation in the paper might not be sufficiently clear. To clarify, there are two aspects to consider: Firstly, at the boundaries of the point cloud, there might be missing content that isn't captured by the scanning process. Consequently, we choose not to label these instances as anomalies, as they result from the limitations of the data collection process rather than indicating true anomalies. Secondly, in our dataset, anomalies do not occur at the edges. This absence of edge anomalies doesn't affect the evaluation. In practical scenarios, we believe that if an edge exception goes unnoticed, it's unlikely that we would be able to observe it. Therefore, this particular scenario doesn't require special consideration. In real-world detection scenarios, the process often involves multiple steps, with different camera positions capturing items from various angles. Any missing information at the edge of one camera position is likely to be captured by another position. Assuming that there is a missing position at this position class exception, there must be a place to check it out.
> > >
> > > ***Q5: I am not fully convinced by the reasoning the authors provide for the single-view test samples ("Since we desire to follow the real-world application: the workers in the production line randomly check one side of the product to identify the defects by matching the scanned data with the prototype."). In their introduction of the dataset, the authors stress the absence of blind spots as an advantage compared to other datasets. As that is only true for the four training samples per object class, this advantage is somewhat limited.***
> > >
> > > The advantage of lacking blind spots, as we mentioned, is primarily noticeable in the training set. This advantage is closely linked to our few-sample scenarios. It is precisely because the training set is devoid of blind spots that the training samples encompass more comprehensive information. In theory, each training sample can effectively capture the complete appearance and shape of a normal sample. This foundation enables us to conduct reliable anomaly detection within few-shot scenarios. If the test set were also devoid of blind spots, it would indeed yield greater consistency with the training set and diminish the complexity of anomaly detection. However, this configuration does not align with real-world practicality. In practice, combining multiple scan results is exceedingly time-consuming and often necessitates manual intervention. This approach contradicts the swift and automated nature of production line scenarios. Hence, by amalgamating considerations from academic research and practical application, we have chosen to maintain a training set without blind spots, while implementing single-sided scanning for the test set.

---

> > > > ### Author Response · Authors · 2023-08-13
> > > > **Response to reviewer 6MP5(Part 4)**
> > > >
> > > > ***Q6: I think formula (1) is incorrect. Should the argmin over $m^l$  be a min? In the definition of $s^{l\*}$, it is unclear what $m^{test}$ is.
> > > > Thanks for the reviewer's correction.***
> > > >
> > > > We have revised this typo in the updated manuscript, which is described below.
> > > >
> > > > $m^{test,\*},m^{l\*} = \underset{m^{test} \in \mathcal{P}(x^{test})}{\arg max} \underset{m^{l} \in \mathcal{M}^{l}}{\arg min}\left\| m^{test} - m^{l}\right\|_{2} \ \ , \quad s^{l\*} = |m^{test,\*} - m^{l\*}|_2  $
> > > >
> > > > ***Q7: For example, in line 181, the authors talk about the "mean proportion of abnormal point clouds". From this sentence alone, I would think that this proportion is 0.5, as there are (roughly) 50 normal and 50 abnormal points clouds per object category. However, I think it refers to the "Anomaly Ratio" in Table 2, which I assume is the mean ratio of points labelled as "abnormal" in a point cloud.***
> > > >
> > > > Thanks for the reviewer correction and sorry for making you confused. We have rectified these typos in the updated manuscript. Specifically, we have revised *"mean proportion of abnormal point clouds"* to *"the mean proportion of abnormal points"* and *"forty thousand to millions point clouds for each item"* to *"forty thousand to million points for each item"*.
> > > >
> > > > ***Q8: In line 182, the authors mention that Table 2 shows "the minimum, maximum, and the average number of point clouds in both the training and test datasets". I do not know which part of the table this refers to.***
> > > >
> > > > Sorry for making you confusing. We have deleted this sentence in the updated manuscript.
> > > >
> > > > ***Q9: At several points in the paper, the word "point cloud" is used where I think "points" is meant. For example, in the abstract the authors mention "forty thousand to millions point clouds for each item". Should this be "forty thousand to million points per cloud"?***
> > > >
> > > > Thank for the reviewer correction. We have revised this in the updated manuscript.
> > > >
> > > > ***Q10: How are missing parts of the objects labelled in the point clouds? As the 3D scanner does not provide any points in those regions, it is not clear what one could annotate. Introducing new points labelled as abnormal is problematic in my opintion, as there are no defined coordinates for missing points.***
> > > >
> > > > Thanks fo the reviewer insightful suggestion. Labeling missing points in 3D objects is indeed a challenging issue. To address this, we have drawn inspiration from the MVTec 3D-AD dataset, which designates the points surrounding the absent region as abnormal. For instance, if a carrot is broken and a section of the middle is missing, the broken cross-section and edges would be labeled as abnormal. In these cases, we do not introduce additional point clouds in the annotation process to account for incompleteness. To provide further clarification, we have incorporated a more detailed explanation in Figure 5 of the updated manuscript.
> > > >
> > > > ***Q11: I found a few typos, e.g., in line 97 "The abnormal images contain 1,148 and ...". I think a word is missing after "1,148".***
> > > >
> > > > Thanks for the reviewer correction. We have revised this typo in the updated manuscript, "The dataset comprises a total of 41 distinct types of anomalies, with a combined count of 1148 anomaly regions."

---

> ### Author Response · Authors · 2023-08-17
> **Further Discussion**
>
> Dear Reviewer 6MP5:
>
> Thank you for taking the time to review our work. We believe that our rebuttal has addressed your questions and concerns. We would be more than happy to discuss with you if you have any further questions about the paper or our rebuttal.
>
> Best,
>
> The authors of Real3D-AD

---

> > ### Author Response · Authors · 2023-08-23
> >
> > Dear Reviewer 6MP5:
> >
> > Since the discussion period is approaching the **deadline (29th August)** and there are only **6 days** left, could you help us to give the feedback for our rebuttal? So in that case, we can have more discussion if you still has any concerns about Real3D-AD. We are eager to achieve your valuable advice to improve the manuscript.

---

> > > ### Comment · Reviewer_6MP5 · 2023-08-23
> > > **Response to the authors**
> > >
> > > Thank you very much for your response which addressed a lot of my concerns. I apologize for the delay in my response.
> > >
> > > **@ Reply to Q1:**
> > >
> > > Thank you for providing the comparison to MVTec 3D-AD. For the rigid objects in that dataset, I agree that the type of anomalies is similar to the objects and anomalies in your dataset. I would argue that incompleteness and redundancy for objects with variations (e.g., "peach") pose a more difficult challenge. For rigid objects, there is more or less one object shape that defines anomaly-free objects. For non-rigid objects, that is not possible. I agree with you that your benchmark results show that the proposed dataset indeed poses a significant challenge. However, I still think that a more diverse range of object types and associated anomalies could have enhanced the dataset (which is also true for the MVTec 3D-AD dataset).
> > >
> > > **@ Reply to Q2:**
> > >
> > > Thank you very much for the additional results with more prototypes. That is a very interesting result. This addresses my concern sufficiently.
> > >
> > > **@ Reply to Q3 and Q4:**
> > >
> > > Thank you for the reply. This addresses my concern sufficiently.
> > >
> > > **@ Reply to Q5:**
> > >
> > > Thank you for the explanation. I understand your reasoning concerning real-world applicability. This addresses my concern sufficiently.
> > >
> > > **@ Reply to Q6:**
> > >
> > > I still find the double argmax/argmin a bit confusing (or I might just not be familiar with this notation). If I understand it correctly, it is supposed to denote the following:
> > >
> > > $$m^{test,\*} = \underset{m^{test}\in\mathcal{P}(x^{test})}{\arg\max}\min_{m^l\in\mathcal{M}^l}\||m^{test}-m^l\||_2\ ,\ m^{l\*}=\underset{m^l\in\mathcal{M}^l}{\arg\min}\||m^{test,\*}-m^l\||_2$$
> > >
> > > Is my understanding correct?
> > >
> > >
> > > **@ Reply to Q7, Q8, Q9:**
> > >
> > > Thank you for the clarification.
> > >
> > > **@ Reply to Q10:**
> > >
> > > Thank you for the explanation. I think the way you labelled the missing parts makes sense.
> > >
> > > A lot of my concerns have been addressed, which is why I am going to increase my rating.

---

> > > > ### Author Response · Authors · 2023-08-23
> > > > **Thasnks Reviewer 6MP5 for your support for our work**
> > > >
> > > > Dear Reviewer 6MP5,
> > > >
> > > > Thank you very much for your positive feedback and increasing your rating.
> > > >
> > > > We agree with your perspective on non-rigid objects. Detecting anomalies in non-rigid objects is indeed a more challenging task, and it will be an important direction for our future extended work.
> > > >
> > > > Regarding formulas about $m^{test,\*}$ and $m^{l\*}$, our wording did cause some confusion. Your understanding of the formulas is correct, and we have revised the manuscript according to your suggestions. We greatly appreciate your suggestion.
> > > >
> > > > Once again, thank you very much for your thorough review and valuable suggestions. Your insights have significantly contributed to improving the paper and inspired us toward potential future research direction.

---

### Official Review · Reviewer_TiJd · 2023-07-19
**Review comments**

**Rating:** 7
**Confidence:** 4
**Correctness:** Yes, the claims seems correct to me.
**Clarity:** Yes, a well written paper.

**Strengths:**

1. Unlike previous datasets for 3D anomaly detection that were either RGB-D or synthetic, this dataset consists of authentic 3D point clouds generated by a 3D scanner. This dataset exhibits a significant improvement in both quality and category diversity, which has the potential to drive advancements in this field.

2. The benchmark and the toolkit promote the fair evaluation for future research.

**Additional Feedback:**

Please refer to the Opportunities For Improvement section for my feedback. Thanks.

**Documentation:**

The paper presents data collection and annotation process. Documentation is good in general.

**Ethics:**

No ethics issue for this dataset.

**Limitations:**

No discussion on limitations and negative cosmical impact in the paper. One potential limitation of the dataset is that adversaries may use the point could models to create counterfeit corresponding manufacturing toys.

**Opportunities For Improvement:**

1. Real3D-AD appears to primarily consist of point cloud representations of 3D manufacturing toys. It would be beneficial to enhance the dataset by incorporating a wider range of objects commonly encountered in human daily life.

2. In figure 5, there appears to be a notable disparity between the point cloud numbers in the training samples and the normal/abnormal samples in the test set. It is strongly recommended to address or provide an explanation for this observation in the form of a discussion.

3. My understanding of the Real3D-AD data are generated by by a 3D scanner. But in line 219-220 of the paper, it states that the test samples is scanned on only one side. This statement is quite confusing. Please clarify this statement.

**Relation To Prior Work:**

Yes. The difference between this study and prior works are specified.

**Summary And Contributions:**

This paper paper introduces Real3D-AD, a dataset designed for 3D point cloud anomaly detection. Real3D-AD comprises 12 categories of 3D models, with each category having 4 normal 3D point cloud prototypes captured in high sensing resolution. The authors conducted an experimental study to assess previous approaches using Real3D-AD and provided a benchmark along with a toolkit for algorithm evaluation. The significance of this work lies in its potential contribution to the 3D point cloud anomaly detection.

---

> ### Author Response · Authors · 2023-08-13
> **Response to reviewer TiJd**
>
> We sincerely thank you for your appreciation of our work **"a significant improvement in both quality and category"**, **"has the potential to drive advancements in this field"**, **"promote the fair evaluation for future research"**, **"a well written paper"**. Next, we respond to your concerns in a point-by-point manner as follows.
>
> ***Q1: Real3D-AD appears to primarily consist of point cloud representations of 3D manufacturing toys. It would be beneficial to enhance the dataset by incorporating a wider range of objects commonly encountered in human daily life***
>
> Thanks for the reviewer valuable suggestion. It is indeed a good idea to include a diverse set of objects commonly encountered in human daily life. However, the primary objective of Real3D-AD is to evaluate the performance of existing 3D-AD methods. We have observed that a significant number of these methods perform poorly on Real3D-AD, as indicated by the ADBench-3D scores (refer to Table 4 in the manuscript). Consequently, we consider Real3D-AD to be a challenging dataset for point cloud anomaly detection.
>
> ***Q2: In Figure 5, there appears to be a notable disparity between the point cloud numbers in the training samples and the normal/abnormal samples in the test set. It is strongly recommended to address or provide an explanation for this observation in the form of a discussion.***
>
> We appreciate the reviewer's observation. The purposed setting in Real3D-AD (as outlined in Section 4.1 of the manuscript) results in a significant contrast between the number of point clouds in the training samples and the count of normal/abnormal samples within the test set. To elaborate further, the training samples offer complete prototypes of the 3D objects, while the test samples are obtained from scans taken on a single side. As a consequence, the number of point clouds of training samples greatly exceeds that of test samples. In order to provide additional clarity on this matter, we have included a more comprehensive analysis in Section 3.2, which reads: "In specific, the training samples provide full prototypes of 3D objects while the test samples are scanned on only one side. Consequently, the number of training samples is considerably larger than that of test samples."
>
> ***Q3: My understanding of the Real3D-AD data is generated by a 3D scanner. But in line 219-220 of the paper, it states that the test samples are scanned on only one side. This statement is quite confusing. Please clarify this statement.***
>
> Thanks for the reviewer valuable advice. The reason why each test samples is scanned on only one side is described in Section 4.1, **"To simulate a real-world application, the test sample is scanned on only one side. Since we desire to follow the real world application: the workers or quality inspection equipments in the production line randomly check one side of the product to identify the defects by matching the scanned data with the prototype."**. As for more clear demonstration, we have highlighted this part in Section 4.1.
>
> ***Q4: No discussion on limitations and negative cosmical impact in the paper. One potential limitation of the dataset is that adversaries may use the point could models to create counterfeit corresponding manufacturing toys.***
>
> Due to the length constraints of the main text, our discussion of limitations and potential negative effects is presented in the supplementary material. In light of the matter related to the potential reverse production of toys, I would like to extend my gratitude to the esteemed reviewer for his/her meticulous examination. While we have indeed provided the results of point cloud scans for the toys, it's crucial to acknowledge that there exists a noticeable disparity between the required 3D template essential for toy fabrication and the results obtained from our point cloud scanning process. As a result, the process of reverse-engineering the toy remains unattainable. Furthermore, it's important to emphasize that the critical factor in the domain of toy manufacturing lies not solely in the template itself but rather in the precise execution of the material process. Therefore, it should alleviate the concerns regarding the potential for reverse engineering.

---

> ### Author Response · Authors · 2023-08-17
> **Further Discussion?**
>
> Dear Reviewer TiJd:
>
> Thank you for taking the time to review our work. We have provided corresponding responses, which we believe have covered your questions and concerns. We want to further discuss with you whether or not your concerns have been addressed.
>
> Best,
>
> The authors of Real3D-AD

---

> > ### Comment · Reviewer_TiJd · 2023-08-21
> > **Thanks for the feedback from the authors.**
> >
> > The rebuttal addresses my concerns. Thank you.

---

> > > ### Author Response · Authors · 2023-08-21
> > > **Thasnks Reviewer TiJd for your support for our work**
> > >
> > > Dear Reviewer TiJd,
> > >
> > > We're delighted that we could address your concerns. Thank you for your recognition of our paper and your valuable suggestions.

---

### Official Review · Reviewer_vFUK · 2023-07-24

**Rating:** 7
**Confidence:** 2
**Correctness:** Please see the Opportunities For Impr…
**Clarity:** Please see the Opportunities For Impr…

**Strengths:**

1. The paper has a clear motivation, which is the 2.5D to 3D gap. Previous datasets uses RGB-D images, which do not capture the whole object and leave "blind spot". The proposed dataset Real3D-AD closes the gap by collecting 1,254 high accuracy full object scans. The full-observation training to partial-observation testing setting also fits in real-world applications.
2. The paper provides a systematic benchmark, ADBench-3D, to various exisitng 3D anomaly detection approach.
3. The paper proposed an effective approach, Reg3D-AD by first aligning the test sample to prototypes via RANSAC and then computing the similarity to local and global features banks. The approach is simple and achieves the best performance on various object categories.

**Additional Feedback:**

Please see the Opportunities For Improvement section.

**Documentation:**

Please see the Opportunities For Improvement section.

**Ethics:**

No.

**Limitations:**

Yes.

**Opportunities For Improvement:**

1. The source of the anomoly is not clear. Is it from the objects or the hardware, and how general are the collected anomalies? The authors mention that they have two types of anomalies including incompleteness and reduncdancy, are they comprehensive or are they representative to all types of anomalies?
2. The description of the objects is not clear. For example, are all described objects "toys" since "Diamond" and "Duck" have similar sizes?  And why are most classes transparent?
3. It is not clear how objects in a class are distributed. It would be better if there is a description or a visualization on that.
4. The authors mention that "There are certain differences between the test set and the training set samples, which need to be addressed." Will this be a limiation to the registration-based approach, since an accurate registration might be hard to get when the testing samples are very different from training samples?

Minor:
1. L227: typo, 3ADBench -> ADBench

**Relation To Prior Work:**

Please see the Opportunities For Improvement section.

**Summary And Contributions:**

Real3D-AD collects a datasets for point cloud anomaly detection. Existing 3D anomaly detection approaches have been benchmarked, and a new registration-based approach has been proposed.

---

> ### Author Response · Authors · 2023-08-13
> **Response to reviewer vfUK(Part 1)**
>
> We sincerely thank you for the detailed comments and positive feedback such as **"clear motivation"**, **"systematic approach"** and **"effective approach"**. For each detailed question, we provide reponses below.
>
> ***Q1: The source of the anomaly is not clear. Is it from the objects or the hardward, and how general are the collected anomalies***
>
> The anomalies are artificial. Due to confidential business information, most of the real abnormal products are restricted from being made public. However, the anomaly types of Real3D-AD are consistent with those of the real anomalies. Specifically, as shown in Table 1 of Q2, we have compiled a comprehensive statistics table of anomaly types from publicly available datasets. Based on this table, we have determined that Real3D-AD predominantly consists of various types of anomalies. Hence, even though the anomalies in Real3D-AD are artificial, it still serves as a highly representative dataset for point cloud anomaly detection.
>
> ***Q2: The authors mention that they have two types anomalies including incompleteness and redundancy, are they comprehensive or are they representative to all types of anomalies?***
>
> The anomaly types showed by Real3D-AD are representative of all categories of anomalies. The primary objective of Real3D-AD is to assess the efficacy of point cloud anomaly detection. Therefore, the anomalies in Real3D-AD are defined based on the quality and shape of point clouds, specifically focusing on the incompleteness and redundancy of the point cloud. Furthermore, the anomaly types in Real3D-AD exhibit consistency with the anomaly types found in RGBD datasets. The comparison is carried out between the Real3D-AD and MVTec 3D-AD datasets, both of which are widely recognized as some of the most popular 3D anomaly detection datasets. A statistics table, designated as Table 1, is constructed to present the various types of anomalies. The anomalies observed in MVTec 3D-AD[1] can be categorized into two types: incompleteness and redundancy, as indicated in Table 1. The first row in Table 1 represents the object categories, and each column corresponds to a specific anomaly category of the object. For instance, in the case of bagels, four distinct categories of anomalies can occur: crack, hole, contamination, and combined. Cracks and holes are classified within the category of incompleteness, while contamination is categorized as a form of redundancy. The "combined" category encompasses both incompleteness and redundancy. The majority of anomaly types listed in Table 1 can be classified as instances of incompleteness and redundancy, with the exception of the color anomaly. Therefore, we firmly assert that the anomaly types demonstrated by Real3D-AD are representative of all anomaly types.
>
> ***Table 1: The statistics of anomaly types in MVTec 3D-AD[1].***
>
>
>
> |   anomaly type\object type                             | bagel         | cable_gland | carrot        | cookie        | dowel         | foam          | peach         | potato        | rope          | tire          |
> |--------------------------------|---------------|-------------|---------------|---------------|---------------|---------------|---------------|---------------|---------------|---------------|
> | incompleteness                 | crack         | bent        | crack         | crack         | bent          | cut           | cut           | cut           | cut           | cut           |
> | incompleteness                 | hole          | cut         | cut           | hole          | cut           |               | hole          | hole          | open          | hole          |
> | incompleteness                 |               | hole        | hole          |               |               |               |               |               |               |               |
> | incompleteness                 |               | thread      |               |               |               |               |               |               |               |               |
> | reduncdancy                    | contamination |             | contamination | contamination | contamination | contamination | contamination | contamination | contamination | contamination |
> | incompleteness and reduncdancy | combined      |             | combined      | combined      | combined      | combined      | combined      | combined      | combined      | combined      |
> | not included                   |               |             |               |               |               | color         |               |               |               |               |
> |

---

> > ### Author Response · Authors · 2023-08-13
> > **Response to reviewer vfUK(Part 2)**
> >
> > ***Q3: The description of the objects is not clear. For example, are all described objects "toys" since "Diamond" and "Duck" have similar sizes?***
> >
> > Yes, "Diamond" and "Duck" have similar sizes. The details of each object are described in Table 2 of the manuscript, which includes information about the size, attributes, number of training samples, and number of test samples. The objects presented in Real3D-AD are toys sourced from manufacturing lines. Thank you for your suggestion. We have provided further clarification about the objects by stating, "These categories include but are not limited to Airplane, Candybar, Chicken, Diamond, Duck, Fish, Gemstone, Seahorse, Shell, Starfish, and Toffees. All of these categories consist of toys derived from manufacturing lines."
> >
> > ***Q4: Why are most classes transparent?***
> >
> > Unlike MVTec 3D-AD[1] and Eyecandies[2], the motivation behind Real3D-AD is to evaluate the performance of point cloud-based anomaly detection, specifically focusing on **3D > 2.5D**, as stated in the manuscript. The anomalies associated with transparent objects pose challenges when utilizing RGB features for identification, but they are relatively easier to detect using point cloud features. So that is why most of classes are transparent in Real3D-AD.
> >
> > ***Q5: The authors mention that "There are certain differences between the test set and the training set samples, which need to be addressed." Will this be a limitation to the registration-based approach, since an accurate registration might be hard to get when the testing samples are very different from training samples?***
> >
> > Yes, the distinction between the samples in the test set and the training set arises from the limitations of the registration-based approach. As illustrated in Table 4 of the manuscript, the baseline method we propose employs a straightforward point cloud registration technique (RANSAC) and achieves an object-level AUROC score of 70.4. Therefore, there is ongoing consideration for enhancing the baseline method's performance in the future.
> >
> > ***Q6: Minor: L227: typo, 3ADBench -> ADBench-3D***
> >
> > Thanks for your corrections. We have revised this typo in the manuscript.
> >
> > [1] Paul Bergmann, Xin Jin, David Sattlegger, Carsten Steger: The MVTec 3D-AD Dataset for Unsupervised 3D Anomaly Detection and Localization; Proceedings of the 17th International Joint Conference on Computer Vision, Imaging and Computer Graphics Theory and Applications - Volume 5: VISAPP, 202-213, 2022, DOI: 10.5220/0010865000003124.
> >
> > [2] Bonfiglioli, Luca, Marco Toschi, Davide Silvestri, Nicola Fioraio, and Daniele De Gregorio. "The eyecandies dataset for unsupervised multimodal anomaly detection and localization." In Proceedings of the Asian Conference on Computer Vision, pp. 3586-3602. 2022.

---

> > > ### Author Response · Authors · 2023-08-17
> > > **Questions Remaining?**
> > >
> > > Dear Reviewer vfUK:
> > >
> > > Do you have any remaining questions or concerns following our response? Please let us know. We’d be very happy to do anything we can that would be helpful in the time remaining!

---

> > > > ### Comment · Reviewer_vFUK · 2023-08-20
> > > > **Thanks for the response**
> > > >
> > > > Thank you for the detailed response, it clearly clarifies all my concerns. All missing details have been added to the revised manuscript. Therefore, I would like to raise my score.

---

> > > > > ### Author Response · Authors · 2023-08-20
> > > > > **Thasnks Reviewer vFUK for your support for our work**
> > > > >
> > > > > Dear Reviewer vFUK,
> > > > >
> > > > > Thank you very much for your positive feedback and for raising your score. Once again, we greatly appreciate your diligent review and the valuable insights you've provided for the paper.

---

### Author Response · Authors · 2023-08-13
**Global Response**

We thank all reviewers for their careful reading and considerate feedback.

We are glad that reviewers agree that Real 3D-AD dataset is a novel and valuable dataset, which has a potential to drive advancements in 3D anomaly detection(**"has a clear motivation"**(vFUK), **"exhibits a significant improvement in both quality and category diversity, which has the potential to drive advancements in this field"**(TiJd), **"is a valuable contribution to the field"**(6MP5), **"presents a novel promising scene for 3D anomaly detection"**(KAeA), **"exhibits the possibility of emerging as a novel paradigm in the realm of 3D anomaly detection"**(KAeA)). We are further glad that reviewers agree our established benchmark and baseline method are reasonable and valuable(**"achieves the best performance on various object categories"**(vFUK), **"promote the fair evaluation for future research"**(TiJd), **"the introduced method is a sound approach"**(6MP5), **"propose a novel baseline method and suggest potential directions for future improvements"**(KAeA)).

The main concern are i: **whether the test set should be scanned from one point of view** and ii: **whether the anomaly types in the dataset is appropriate**. We thank their attention to these details, and we would like to provide an explanation for our choices in this regard.



**I: Whether the test set should be scanned from one point of view.**

On one hand, setting the test set as single-sided scans aligns with real-world considerations. In practical applications, it is more common to perform single-sided scans of objects using a single machine. On the other hand, if we were to set the test set for 3D anomaly detection to be entirely free of blind spots, it would create a significant disparity with 2D anomaly detection. This divergence might limit the potential for future convergence between the two domains, which contradicts our expectation of a unified paradigm encompassing both 2D and 3D anomaly detection. In essence, we aim to establish a bridge between industrial applications and academic research rather than further dividing them. Therefore, the decision to have a test set consisting of single-sided scans was made after carefully considering these factors.

**II: Whether the anomaly types in the dataset are appropriate**

Our dataset encompasses three types of anomaly samples: i) incompleteness, ii) redundancy, and iii) combined incompleteness and redundancy. These types cover the majority of anomaly cases, providing a comprehensive representation of anomalies. In the case of MVTec 3D-AD[1], 40 out of the 41 different object anomaly types fall within these categories, which further validates the high representativeness of these anomaly types.

Moreover, since our dataset is unsupervised (with the training set containing only normal data), the variety of anomalies in the test set has a relatively minor impact on evaluating the model's performance. Therefore, we believe that the introduced types of anomalies are sufficient to assess the effectiveness of anomaly detection models.

Certainly, we are committed to addressing each of the reviewer's concerns individually. We sincerely thank all reviewers for their time and efforts and appreciate the opportunity to collaborate with them to enhance the quality and comprehensiveness of our work. By collectively addressing these concerns, we aim to refine our research and contribute to the advancement of the field.

[1] Paul Bergmann, Xin Jin, David Sattlegger, Carsten Steger: The MVTec 3D-AD Dataset for Unsupervised 3D Anomaly Detection and Localization; Proceedings of the 17th International Joint Conference on Computer Vision, Imaging and Computer Graphics Theory and Applications - Volume 5: VISAPP, 202-213, 2022, DOI: 10.5220/0010865000003124.

---

### Decision · Program_Chairs · 2023-09-22

**Decision:**

Accept (Poster)

**Comment:**

The authors introduce a new dataset for anomaly detection in 3D point clouds. The dataset was created using a high-precision 3D scanner on real objects, leading to very high-resolution point clouds. Along with the dataset, the authors also provide a full evaluation toolkit that can be used to evaluate anomaly detection methods on the dataset.

Moreover, the authors also introduce a novel method for 3D point cloud anomaly detection.

All the reviewers are positive about this contribution.